# Three Novel Zn-Based Coordination Polymers: Synthesis, Structure, and Effective Detection of Al^3+^ and S^2−^ Ions

**DOI:** 10.3390/molecules25020382

**Published:** 2020-01-17

**Authors:** Yuna Wang, Xiaofeng Zhang, Yanru Zhao, Suoshu Zhang, Shifen Li, Lei Jia, Lin Du, Qihua Zhao

**Affiliations:** 1School of Chemical Science and Technology Pharmacy, Yunnan University, Kunming 650091, China; wyn123456LL@163.com (Y.W.); zxf123456LL@163.com (X.Z.); zyr123456LL@163.com (Y.Z.); zss123456@163.com (S.Z.); lsf123456LL@163.com (S.L.); jl123456LLL@163.com (L.J.); 2Key Laboratory of Medicinal Chemistry for Natural Resource Education Ministry, Yunnan University, Kunming 650091, China

**Keywords:** coordination polymers, 4-methoxyisophthalic acid, ion detection, ethanol

## Abstract

Three novel Zn-based coordination polymers (CPs), [Zn(MIPA)]_n_ (**1**), {[Zn(MIPA)(4,4′-bipy)_0.5_(H_2_O)]·1.5H_2_O}_n_ (**2**), and {[Zn(MIPA)(bpe)]·H_2_O}_n_ (**3**) (MIPA = 4-methoxyisophthalic acid, 4,4′-bipy = 4,4′-bipyridine, bpe = (E)-1,2-di(pyridine-4-yl)ethane), were constructed by ligand 4-methoxyisophthalic acid under solvothermal conditions. Compound **1** features a beaded 2D-layer architecture, while compound **2** presents a 2-fold interpenetrating structure with a uninodal three-connected **hcb** topology. Compound **3** has a 3-fold interpenetrated four-connected **dmp** topology. Photoluminescence investigations of compound **2** were explored in detail, by which ions were detected, and it was observed to have the highest quenching efficiency toward Al^3+^ and S^2−^ ions. The possible fluorescence quenching mechanisms of **2** toward Al^3+^ and S^2−^ ions were also explored. To the best of our knowledge, this is the first potential dual-responsive luminescent probe based on a Zn(II) coordination polymer for detecting Al^3+^ and S^2−^ ions via a luminescence quenching effect in ethanol.

## 1. Introduction

As interest in environmental and health concepts grows, the effects of various ions on the biological environment and human health are popular focus points. Aluminum and sulfur ions play an important part in our daily lives and are not able to be substituted due to their peculiarly superior properties, such as the light weight and corrosion resistance of the metal aluminum, and the participation of sulfur ions in physiological processes [1,2]. However, if the aluminum ion or sulfur ion content in the body is high, it will affect human health. The concentration of aluminum ions accumulated in the human body can result in metabolic disorders. Abnormal production of sulfur ions is associated with many diseases, including loss of consciousness and respiratory paralysis [3,4]. In addition, sulfur ions are important pollutants in the environment and are widely found in various polluted water sources. Therefore, it is very important to urgently establish a rapid and effective analytical method for the determination of aluminum and sulfur ions or to find a substance that has a specific and sensitive ability to recognize them.

Coordination polymers (CPs) have aroused much interest, owing to their potential applications in catalysis [5,6,7], adsorption [8,9], separation [10], electrochemistry [11,12,13], and as chemical sensors [14,15,16,17]. In recent years, coordination polymers (CPs) used as fluorescent sensors have been successfully applied in the determination of various analytes such as metal ions (Fe^3+^, Cr_2_O_4_^2−^, etc.) [18,19,20], small molecules (acetylacetone, water, etc.) [21,22], and explosives (TNT, PA, etc.) [23,24,25]. Coordination polymers have become one of the hotspots of material chemistry as new fluorescent probes because of their high sensitivity and selectivity, ease of operation, convenience, and fast response time [26,27,28,29]. There has been evidence that Zn-based coordination polymers present prominent superiority, especially in the field of fluorescence sensing [30,31]. He et al. designed a Zn-based multi-responsive luminescent probe with a discriminatory ability for Fe^3+^, Al^3+^, SiF_6_^2−^, Cr_2_O_7_^2−^, nitrofurantoin (NFT), and nitrofurazone (NFZ) via the change of luminescence intensity [32]. The Zn(II) ion, as a d^10^ metal ion, can greatly enhance the rigidity and fluorescent yield of coordination polymers [33,34]. Furthermore, Zn-based CPs are able to display the ligand-centered characteristic emission. As such, Zn-based CPs used as fluorescent sensors usually change the fluorescence properties by means of the incorporation, functionalization, and modification of the ligands [35,36]. Therefore, it is fascinating and significant to design and synthesize luminescent materials using Zn(II) ions and the control of ligands.

Herein, we reported three novel Zn-based coordination polymers: [Zn(MIPA)]_n_ (**1**), {[Zn(MIPA)(4,4′-bipy)_0.5_(H_2_O)]·1.5H_2_O}_n_ (**2**), and {[Zn(MIPA)(bpe)]·H_2_O}_n_ (**3**) (MIPA = 4-methoxyisophthalic acid, bpe = (E)-1,2-di(pyridine-4-yl)ethane), which were constructed by introducing an N-donor ligand and controlling solvents. The solid fluorescence properties of compounds **1**–**3** were investigated, and results clearly indicated that compound **2** has a good fluorescence performance in Al^3+^ and S^2−^ detection, with high selectivity and sensitivity. As far as we know, this is the first Zn-based coordination polymer which can be used as a dual functional fluorescent material to detect Al^3+^ and S^2−^ ions by means of luminescence quenching in ethanol.

## 2. Results

### 2.1. Crystal Structure of [Zn(MIPA)]_n_
*(**1**)*

Single crystal X-ray analysis showed that compound **1** crystallizes in the orthorhombic Pbca space group, presenting a 2D multi-metallic linear cluster structure. The asymmetrical unit consists of one Zn(II) ion and one MIPA ligand. Each hexacoordinated Zn(II) center with a ZnO_6_ binding set displays a distorted octahedral geometry, created by five carboxylic O atoms from four different MIPA ligands (Zn–O: 1.958(4), 1.997(4), 2.008(4), 2.016(4) Å) and one methoxy O atom (Zn–O: 2.482(4) Å) (Figure 1a). The MIPA ligand is completely deprotonated with the coordination mode of μ_4_-κ^5^O′,O″,O‴,O′′′′:O′′′′′. (Scheme 1, mode I). Two carboxyl groups from different MIPA ligands are used to bridge the neighboring Zn(II) ions into 1D multi-metallic (Zn–O)_n_ chains (Zn···Zn distance is 3.519 Å), and the neighboring 1D multi-metallic chains are further connected into a beaded 2D-layer structure by MIPA ligands (Figure 1b,c). The 2D-layer structure is further assembled into a 3D supermolecule frame by π···π interaction between layers (Figure 1d).

### 2.2. Crystal Structure of {[Zn(MIPA)(4,4′-bipy)_0.5_(H_2_O)]·1.5H_2_O}_n_
*(**2**)*

Single crystal X-ray analysis revealed that **2** crystallizes in the monoclinic *C*2/*c* space group. The asymmetric unit is composed of one Zn(II) ion, one MIPA, half of a 4,4′-bipy linker, one coordinated water molecule, and one-and-a-half free water molecules. The Zn1 unit is four-ligated with distorted tetrahedral ZnO_3_N geometry by two carboxylic O atoms (Zn–O: 1.991(3), 1.948(3) Å) from two MIPA ligands, one coordinated water molecule (Zn–O: 2.055(3) Å), and one N atom (Zn–N: 2.024(4) Å) from a 4,4′-bipy linker (Figure 2). The fully deprotonated MIPA ligands connect the neighboring Zn (II) ions into a 1D W-type chain. As shown in Scheme 1, mode II, all the carboxylic groups of MIPA ligands present a *syn-syn-μ*_2_-*κ*^2^O,O′ coordination mode. The 1D chains are arranged in a parallel mode by the 4,4′-bipy linkers, which results in a zigzag 2D hexagone-pattern fold (Figure 3a,b). The terminally-coordinated water molecules that are suspended between the two layers prevent its further extension to a high-dimensional network. Two of the folds are interlocked with each other to give a 2-fold interpenetrating structure (Figure 3c). To better observe the structure, topology analysis was carried out. The Zn(II) ions can be considered as three-connected nodes, while all the MIPA and bpe ligands can be viewed as linkers. Therefore, the structure of compound **2** can be simplified as a uninodal three-connected **hcb** topology with the Schläfli symbol of 6^3^ (Figure 3d).

### 2.3. Crystal Structure of {[Zn(MIPA)(bpe)]·H_2_O}_n_
*(**3**)*

Compound **3** crystallizes in the orthorhombic space group *Pnna*, presenting a 3-fold interpenetrating 3D structure. An asymmetric unit contains half of a Zn(II) ion, half of an MIPA ligand, half of a bpe linker, and half of a free water molecule. The Zn(II) ion exhibits distorted tetrahedral ZnO_2_N_2_ geometry as it coordinates with two carboxylic O atoms (Zn–O: 1.934(3) Å) from two different MIPA ligands, and two pyridyl N atoms (Zn–N: 2.059(4) Å) from different bpe ligands (Figure 4a). MIPA ligands bind the two adjacent Zn(II) ions resulting in the formation of a 1D chain (Figure 4b). Here, each carboxylate group adopts a *syn-syn-μ*_2_-*κ*^2^O,O′ coordination (Scheme 1, mode II). The 1D helical chain further links with four helices in opposite directions through the four coplanar bpe ligands from the view of the *b* axis. Consequently, it is extended into a 3D porous framework (Figure 4c). Moreover, three identical frameworks interpenetrate with each other to stabilize the whole structure and almost block the pores. The two-connected bridging of the MIPA ligand and the bpe ligand can be viewed as a linear link. The Zn(II) ions can be regarded as four-connected nodes. Therefore, the overall structure of **3** is simplified as a uninodal four-connected 3-fold interpenetrating 3D structure with a point symbol of (6^5^·8) (Figure 4d).

### 2.4. Powder Diffraction (PXRD) Analysis and Thermogravimetric Analysis (TGA)

For compounds **1**–**3**, bulk phase purity was confirmed by the PXRD pattern of the simulated and as-synthesized compounds. The results are shown in Appendix A—it is clearly observable that the bulk phase is in good agreement with the simulated data, confirming all of the compounds are high in purity and can support further study. Thermogravimetric analysis (TGA) of compounds **1**–**3** was carried out under an N_2_ atmosphere increasing from 25 to 800 °C with a flow rate of 10 °C·min^−1^. The TG curves are given in Figure 5. For compound **1**, no obvious weight loss before 400 °C was observed, which showed good thermostability. After that, rapid decomposition of the structure occurred. For **2**, a weight loss of 10.4% was detected in the temperature range of 60−120 °C, which corresponds to the release of both free and coordinated water molecules (calculated, 11.77%). Subsequently, no significant weight loss was observed until the structure collapsed at 275 °C. Compound **3** was stable until 320 °C, followed by the decomposition of its backbone.

### 2.5. Solid-State Luminescence

The solid-state luminescent emissions of compounds **1**, **2**, **3**, and free ligands were collected at room temperature. The 4,4′-bpy and bpe ligands displayed luminescent emission at 362 nm and 417 nm, respectively, while the MIPA ligand was not observed to produce luminescent emission (Appendix A). As illustrated in Figure 6a, upon photoexcitation at 323 nm, weak and broad emission of **1** is observed, with a maximum peak of 344 nm (Appendix A). Compound **2** exhibits stronger emissions at 436 nm, with an excitation peak of 346 nm, while compound **3** is not observed to produce any luminescent emission (Appendix A). The luminescence of **1** and **2** is due to ligand-based emission, which may be connected with the metal-to-ligand charge transfer, ligand-to-metal charge transfer, or intraligand transitions between ligands [37,38,39,40]. The photoluminescent emission of CPs is closely associated with the central metal coordination environment and the property of the ligand. After adding and tuning the N-donor linkers, the coordination environment of metal ions and the structure of the coordination polymers are changed. All of those have effects on the rigidity of the structure and further influence the energy transfer and charge transfer involved in luminescence.

### 2.6. Selectivity toward Al^3+^ Ions

Compound **2** was selected for use in studying the luminescence responses to various metal cations, owing to its stronger solid-state emission performance. Before the experiment, the fluorescence spectra of compound **2** in various solvents were compared and studied (Figure 6b). In view of environmental protection and experimental results, ethanol was used as the main solvent. Compound **2** exhibits stronger emissions at 422 nm with an excitation peak of 309 nm in ethanol (Figure 7a). A sample of **2** (1 mg) was ground into powder and dispersed in ethanol solution (4 mL) containing different nitrates (cations: Mn^2+^, Ag^+^, Mg^2+^, Co^2+^, Zn^2+^, Cd^2+^, Pb^2+^, Hg^2+^, Al^3+^) with a concentration of [M] = 0.4 mM. The suspensions were sonicated for 20 min, and then the photoluminescence experiment was conducted. As shown in Figure 7b, with the addition of the different metal cations, most metal cations have different effects on the characteristic peak at 422 nm. Notably, with the addition of Al^3+^ ions, the luminescence intensity at 422 nm substantially decreased. The addition of 0.4 mM Al^3+^ ions can produce a 90% quenching effect. These results indicate that **2** is a powerful sensor for detecting Al^3+^ ions.

To better understand the response toward Al^3+^ ions, the fluorescence spectra upon the additions were collected. As depicted in Figure 8a, with the increasing concentration of Al^3+^ ions, the emission intensity at 422 nm gradually decreased and then disappeared for the suspensions. The Stern−Volmer equation can be used to rationalize the quenching effect: *I*_0_/*I* = 1 + *K*_sv_[M], where *I*_0_ and *I* are the fluorescence intensities before and after adding Al^3+^ ions, respectively; [M] is the molar concentration(mol/L) of Al^3+^ ions; and *K*_sv_ is the Stern−Volmer constant. As expected, the Stern−Volmer curve for the Al^3+^ ion is linearly proportional in the range of 0–133 uM (R^2^ = 0.921), and the value of *K*_sv_ is calculated as 9.61 × 10^3^ M^−1^ (Figure 8b).

To further study the practicality of compound **2** in the waste solution, different metal cations were introduced into the suspension in the same conditions. As shown in Figure 9, most metal ions had little effect on the detection of Al^3+^ ions. These experimental results further indicate that compound **2** is a highly selective luminescent sensor for Al^3+^ ion detection in ethanol.

The PXRDs were recorded after the sensing of metal ions. As confirmed by PXRD patterns (Appendix A), the structure of **2** remains unchanged throughout the sensing process, thus eliminating the collapse of the crystal structure. The Uv-vis absorption spectrum was gained at room temperature (Appendix A). There exists a spectrum overlap between the Uv-vis absorption of Al^3+^ ions and the excitation band of compound **2**, which confirms that there is competitive energy absorption between Al^3+^ ions and compound **2**. As a result, the phenomenon can be explained either by the competitive energy absorption or the combined action of energy-transfer and competitive energy absorption.

### 2.7. Selectivity toward S^2−^ Ions

Given the sensing ability of anions by compound **2**, its ability to sense anions was also examined by the addition of different sodium salts (anions: I^−^, Cl^−^, NO_3_^−^, F^−^, Br^−^, SO_4_^2−^, CH_3_COO^−^, CO_3_^2−^, and S^2−^) in ethanol solutions, respectively, into the ethanol suspensions of compound **2**. As shown in Figure 10, different anions have different degrees of influence on the intensity of luminescence, most notably, S^2−^ ions. When S^2−^ ions are present in solutions, the luminescence intensity exhibits a significant decline. The concentration of 1.75 mM of S^2−^ ions can quench the 90% luminescence intensity. It is clear that **2** is an excellent luminescent sensor for the sensitive and selective detection of S^2−^ ions.

The sensing sensitivity of **2** toward S^2−^ ions was further examined. As depicted in Figure 11a, the titration curves show that the emission intensity of **2** decreased with the addition of increasing concentrations of S^2−^ ions. The luminescence intensity is highly sensitive to S^2−^ ions. The luminescence intensity is observed to be linearly proportional to the concentration of S^2−^ ions in the range of 0−0.333 mM. (R^2^ = 0.969) (Figure 11b), indicating the coexistence of dynamic and static quenching processes. The value *K*_sv_ is estimated to be 2.14 × 10^3^ M^−1^.

To further check the selective quenching behavior of S^2−^ toward **2**, its anti-interference capability was studied. In the same conditions, some other anions were dispersed in suspensions. As shown in Figure 12, most anions had some contribution to luminescence quenching, but the quenching effect grew with the addition of S^2−^ ions. This further proved that compound **2** has remarkable selectivity toward S^2−^ ions in ethanol.

The PXRD pattern of compound **2** after sensing S^2−^ was recorded (Appendix A). The experiment results revealed that the structure of compound **2** remains unchanged throughout the sensing process, demonstrating that the phenomenon is unrelated to the collapse of the frame. The Uv-vis absorption spectrum of S^2−^ ions and the excitation band of compound **2** have an observable spectral overlap, thus competitive absorption cannot be ruled out (Appendix A). As a result, the phenomenon can be explained both by competitive energy absorption or the combined action of energy-transfer and competitive absorption.

## 3. Experimental

### 3.1. Materials and General Methods

All reagents and solvents were purchased commercially and used without further purification. Infrared (IR) spectra were obtained through an FTS-400 FT-IR spectrometer together with a KBr pellet from 4000 to 400 cm^−1^. Powder X-ray diffraction analyses were determined on a Bruker D8 Advance with Cu *Kα* radiation (λ = 1.54 Å). Elemental analyses of C, H, and N were performed in an Elementar Vario EL III analyzer. Thermogravimetric analyses (TGA) were carried out on a Metler-Toledo simultaneous SDT thermal analyzer (Bangkok, Thailand) at a heating rate of 10 °C min^−1^ under a N_2_ atmosphere (N_2_ flow rate = 0.06 L min^−1^). The luminescence spectra were measured on a Lengguang (Shanghai, China) F98 fluorescence spectrophotometer at room temperature.

### 3.2. Synthesis of [Zn(MIPA)]_n_
*(**1**)*

MIPA (4.90 mg, 0.0250 mmol) and Zn(NO_3_)_2_·3H_2_O (29.7 mg, 0.100 mmol) were dissolved in a mixed solvent comprising DMF/H_2_O/CH_3_OH (l/6/8, *v*/*v*/*v*, 2 mL) by sonication, and then sealed in a 5 mL glass vial and kept at 100 °C for 2 days. After being cooled to room temperature over 24 h, colorless slice crystals of **1** were obtained (yield 82% based on MIPA). Elemental analysis calculated (%) for C_9_H_6_O_5_Zn: C 41.65, H 2.31. Found: C 42.09, H 2.33. IR (KBr, cm^−1^): 3436 (vs), 1629 (s), 1537 (m), 1358 (s), 1338 (m), 1001 (w), 791 (w), 636 (w).

### 3.3. Synthesis of {[Zn(MIPA)(4,4′-bpy)_0.5_(H_2_O)]·1.5H_2_O}_n_
*(**2**)*

MIPA (4.90 mg, 0.0250 mmol), 4,4′-bpy (1.95 mg, 0.0125 mmol), and Zn(NO_3_)_2_·3H_2_O (29.7 mg, 0.100 mmol) were dissolved in a mixed solvent comprising DMF/H_2_O (1/2, *v*/*v*, 2 mL) by sonication, and then sealed in a 5 mL glass vial and kept at 100 °C for 2 days. After being cooled to room temperature over 24 h, colorless block crystals of **2** were obtained (yield 67% based on MIPA). Elemental analysis calculated (%) for C_14_H_15_ZnNO_7.5_: C 43.9, H 3.92, N 3.66. Found: C 43.25, H 3.71, N 3.78. IR (KBr, cm^−1^): 3431 (vs), 1619 (s), 1543 (m), 1381 (s), 1267 (m), 1015 (w), 779 (w), 644 (w).

### 3.4. Synthesis of {[Zn(MIPA)(bpe)]·H_2_O}_n_
*(**3**)*

Compound **3** was obtained by the same procedure used for the preparation of **2**, except that 4,4′-bpy was replaced by bpe (1.95 mg, 0.0125 mmol). Colorless block crystals of **3** were obtained (yield 74% based on MIPA). Elemental analysis calculated (%) for C_21_H_17_ZnN_2_O_6_: C 54.9, H 3.70, N 6.10. Found: C 55.05, H 3.66, N 6.36. IR (KBr, cm^−1^): 3502 (s), 1605 (vs), 1500 (w), 1383 (m), 1269 (w), 1021 (s), 779 (s), 680 (w), 551 (m).

### 3.5. Crystallographic Data Collection and Refinement

A Bruker Smart APEX II CCD area-detector was utilized to obtain the crystal data of compounds **1**–**3** at 293 K using ω rotation scans with widths of 0.3° and graphite-monochromated Mo-*Kα* radiation (λ = 0.71073 Å). Empirical absorption corrections were applied using the SADABS program. All structures were solved by the direct method and refined by the full-matrix least-squares method on *F*^2^ with SHELXL-14 software [40,41,42,43]. Non-hydrogen atoms were defined by the Fourier synthesis method. Positional and thermal parameters were refined by the full-matrix least-squares method (on *F*^2^) to convergence. Hydrogen atoms of the ligands were initially localized from different Fourier maps and subsequently placed at geometrically ideal positions using riding models with isotropic displacement parameters derived from their carrier atoms. The hydrogen atoms of water were added by the different Fourier maps and refined with suitable constraints. For compound **3**, the positioning of the methoxy groups is disordered. The pertinent crystallographic details for compounds **1**–**3** are provided in Table 1, while the selected bond lengths and angles are given in Appendix A. Crystallographic data are available at the Cambridge Crystallographic Data Center (see in Appendix B).

## 4. Conclusions

In conclusion, three coordination polymers, [Zn(MIPA)]_n_ (**1**), {[Zn(MIPA)(4,4′-bpy)(H_2_O)_0.5_]·H_2_O}_n_ (**2**), and {[Zn(MIPA)(bpe)]·H_2_O}_n_ (**3**) (MIPA = 4-methoxyisophthalic acid, bpe = (E)-1,2-di(pyridine-4-yl)ethene), have been successfully synthesized by adding auxiliary ligands and controlling solvent effects. All the structures of compounds **1**–**3** were characterized by infrared, PXRD, and TGA studies. The fluorescence properties of compounds **1**–**3** were explored, and it was found that **2** has strong fluorescence emission. Fluorescence titration of compound **2** exhibits a selective and highly sensitive quenching efficiency toward Al^3+^ and S^2−^ ions in ethanol, even in the presence of disturbing ions. Either the presence of competitive energy absorption or the combined action of energy-transfer and competitive energy absorption is the main reason for the detection of the Al^3+^ and S^2−^ ions in ethanol. Moreover, this work is meaningful since it proves the potential for detecting Al^3+^ and S^2−^ ions based on coordination polymers.

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
