# Peer review of "Three Novel Zn-Based Coordination Polymers: Synthesis, Structure, and Effective Detection of Al3+ and S2− Ions"

_molecules, 2020, doi:10.3390/molecules25020382_

Round 1
Reviewer 1 Report
This paper describes the syntheses, crystal structures, and luminescent properties of three zinc(II) complexes with 4-methoxyisophthalic acid with or without bis-pyridyl spacers. The complex containing 4,4’-bipyridine as a spacer (compound 2) shows dual sensing ability aluminum(III) cation and sulfide anion. This is a carefully done study and the findings are interesting. Thus, this paper is worth publishing in Molecules with minor revisions. Additional comments are listed below.
1) line 32: various ions to is --> various ions is
2) line 64: by the means of --> by means of
3) line 67: monoclinic --> orthorhombic
4) line 73, 91 and 115: The expression “μ1” is wrong. There is no such expression. Authors should correct them according to the IUPAC red book.
5) line 90: Authors mention that the zinc ion of compound 2 is four-coordinated. Therefore, this complex does not correspond to “mode II” but “mode III”.
6) line 112: tetrahedral ZnO4N2 --> tetrahedral ZnO2N2, and four carboxylic O atoms --> two carboxylic O atoms
7) line 115: 1D helical structure in Figure 4b is unclear.
8) Authors should mention about the positional disorder of the methoxy group for compound 3.
9) line 146 to 148: The explanations are unclear. In addition, compounds 2 and 3 have the same coordination mode for the ligand (mode III). Author should revise these explanations.
10) Plots in Figure 8b and 11b seem to be a curvature. Authors should reexamine it.
11) Authors should clearly show what indicate unlabeled short bars in both Figure 9 and 12.
12) line 198: Authors should clearly show what salts are employed as anion sources.
Are there any influences by cations?
13) line 220: Al3+ ions --> S2- ion
Author Response
Dear Editor,
Thank you for your kind consideration of this manuscript. We have revised the manuscript according to the reviewers' comments. A detailed response to reviewers' comments is provided. All changes are highlighted in the manuscript.
We hope that you'll find the revised manuscript suitable for publication.
Sincerely yours,
Prof. Qi-Hua Zhao
Response to Reviewer(s)' Comments:
Reviewer 1
line 32: various ions to is --> various ions is
Our Answer: Thanks for the reviewer’s suggestion. We have replaced " various ions to is " to " various ions is ". And the change has been marked in the text.
line 64: by the means of --> by means of
Our Answer: Thanks for the reviewer’s suggestion. We have modified this mistake. And the change has been marked in the text.
line 67: monoclinic --> orthorhombic
Our Answer: Thanks for the reviewer’s suggestion. We have replaced " monoclinic " to " orthorhombic ". And the change has been marked in the text.
line 73, 91 and 115: The expression “μ1” is wrong. There is no such expression. Authors should correct them according to the IUPAC red book.
Our Answer: Thanks for the reviewer’s suggestion. We have corrected these mistakes. And all the changes have been marked in the text. Please see.
line 90: Authors mention that the zinc ion of compound 2 is four-coordinated. Therefore, this complex does not correspond to “mode II” but “mode III”.
Our Answer: Thanks for the reviewer’s suggestion. We have amended these mistakes. And all the changes have been marked in the text. Please see.
line 112: tetrahedral ZnO4N2 --> tetrahedral ZnO2N2, and four carboxylic O atoms --> two carboxylic O atoms.
Our Answer: Thanks for the reviewer’s suggestion. These mistakes have been corrected. And all the changes have been marked in the text. Please see.
line 115: 1D helical structure in Figure 4b is unclear.
Our Answer: Thanks for the reviewer’s suggestion. We are agreed with that 1D helical structure in Figure 4b is unclear. Based on this character, we modified the description of 1D chain. And all the changes have been marked in the text. Please see.
Authors should mention about the positional disorder of the methoxy group for compound 3.
Our Answer: Thanks for the reviewer’s suggestion. We have added the information of the positional disorder of the methoxy group for compound 3 in the part of crystallographic data collection and refinement. And all the changes have been marked in the text. Please see.
line 146 to 148: The explanations are unclear. In addition, compounds 2 and 3 have the same coordination mode for the ligand (mode III). Author should revise these explanations.
Our Answer: Thanks for the reviewer’s suggestion. We have added a short overview of the explanation and give the detailed description in the solid-state luminescence section of the manuscript. And we have corrected this mistake of the mark of coordination mode.
Plots in Figure 8b and 11b seem to be a curvature. Authors should reexamine it.
Our Answer: Thanks for the reviewer’s suggestion. The linear relation between ions concentration and I0/I-1 is not particularly close to 1 in Figure 8b and 11b. However, I find that material of some literature also displays this phenomenon. Take the following content of literature as an example.
Authors should clearly show what indicate unlabeled short bars in both Figure 9 and 12.
Our Answer: Thanks for the reviewer’s suggestion. We have labeled the short bars. And all the changes have been marked in the text. Please see.
line 198: Authors should clearly show what salts are employed as anion sources.
Our Answer: Thanks for the reviewer’s suggestion. We have introduced the sources of anions in line 201. And the change has been marked in the text.
Are there any influences by cations?
Our Answer: Thanks for the reviewer’s suggestion. The cations have effect on the anions detection, but the influence is very small. Considering the solubility of metal salts, the different sodium salts are selected as the resource of anions.
line 220: Al3+ ions --> S2- ion
Our Answer: Thanks for the reviewer’s suggestion. We have replaced " Al3+ ions" to " S2- ions". And the change has been marked in the text.

Reviewer 2 Report
The paper by Qihua Zhao et al. describes Three novel Zn-based coordination polymers based on 4-methoxyisophthalate. These have been prepared via conventional solvothermal conditions and structurally characterized by common methods. The photoluminescence and thermal stability of the compounds presented are also studied. One of the CP has been found to display a good sensing ability to Al3+ cations and S2- anions, which quench the luminescence of the compound. In general, the work seems to be sound and well performed. Thus, I recommend the acceptance of this manuscript after addressing the minor issues listed below.
The possible origin of luminescence of the CPs should be somehow commented. Moreover, if possible, photoluminescence quantum yields for CPs 1–3 should be provided. For completeness, excitation spectra of CPs 2 and 3 should be added in ESI. The preparative yields of 1-3 should be given in the experimental part. The PL intensities have been measured for nine different concentrations of Al3+, meanwhile the Stern−Volmer plot contains only five points (Figure 8). The same true for Figure 11 also. Since the paper deals with luminescent CPs, the following recent papers on luminescent coordination polymers are recommended to add in the citation list: doi: 10.1016/j.inoche.2019.107473, DOI:10.1039/C8QI01302K, doi: 10.1016/j.inoche.2019.107513.Author Response
Dear Editor,
Thank you for your kind consideration of this manuscript. We have revised the manuscript according to the reviewers' comments. A detailed response to reviewers' comments is provided. All changes are highlighted in the manuscript.
We hope that you'll find the revised manuscript suitable for publication.
Sincerely yours,
Prof. Qi-Hua Zhao
Reviewer 2
Comments:
The paper by Qihua Zhao et al. describes Three novel Zn-based coordination polymers based on 4-methoxyisophthalate. These have been prepared via conventional solvothermal conditions and structurally characterized by common methods. The photoluminescence and thermal stability of the compounds presented are also studied. One of the CP has been found to display a good sensing ability to Al3+ cations and S2- anions, which quench the luminescence of the compound. In general, the work seems to be sound and well performed. Thus, I recommend the acceptance of this manuscript after addressing the minor issues listed below.
The possible origin of luminescence of the CPs should be somehow commented.
Our Answer: Thanks for the reviewer’s suggestion. We have added a short overview of the possible origin of luminescence of the CPs and give the detailed description in the solid-state luminescence section of the manuscript. Please see.
Moreover, if possible, photoluminescence quantum yields for CPs 1–3 should be provided.
Our Answer: Thanks for the reviewer’s suggestion. Due to the lack related experimental equipment, photoluminescence quantum yields of compound 1-3 could not be obtained. We have made every effort, but had not been generating satisfying results.
For completeness, excitation spectra of CPs 2 and 3 should be added in ESI.
Our Answer: Thanks for the reviewer’s suggestion. We have added the excitation spectra of 4,4´-bpy, bpe, 1 and 2 in ESI. When I check the graph, I find the luminescence of compound 1 in the manuscript is mistaken for the bpe ligand. And I corrected the Figure 6a. And the change has been marked in the text. For this, I am sorry.
The preparative yields of 1-3 should be given in the experimental part.
Our Answer: Thanks for the reviewer’s suggestion. We have given the preparative yields of compound 1-3 in the experimental part. And the change has been marked in the text. Please see.
The PL intensities have been measured for nine different concentrations of Al3+, meanwhile the Stern−Volmer plot contains only five points (Figure 8). The same true for Figure 11 also.
Our Answer: Thanks for the reviewer’s suggestion. At the beginning, I want to put in more points in the Stern−Volmer polt. But I find that the value of R2 display a greater deviation. For example, if the Stern−Volmer polt contains six points, the value of R2 is only 0.879.
Since the paper deals with luminescent CPs, the following recent papers on luminescent coordination polymers are recommended to add in the citation list: doi: 10.1016/j.inoche.2019.107473, DOI:10.1039/C8QI01302K, doi: 10.1016/j.inoche.2019.107513.
Our Answer: Thanks for the reviewer’s suggestion. We have added a short overview of the aim and novelty of the work and give the detailed description in the introduction section of the manuscript. And we have cited DOI: 10.1016/j.inoche.2019.107473, DOI:10.1039/C8QI01302K and DOI: 10.1016/j.inoche.2019.107513. in refs [34, [37], [38] respectively.
